# Three-Dimensional Acoustic Device for Testing the All-Directional Anisotropic Characteristics of Rock Samples

**DOI:** 10.3390/s22239473

**Published:** 2022-12-04

**Authors:** Kai Zhang, Shengqing Li, Yuanda Su, Baohai Tan, Wenjie Wu, Shoutao Xin

**Affiliations:** 1School of Geosciences and Technology, China University of Petroleum, Qingdao 266555, China; 2Yichang Chengfa Gezhouba Gas Development Corporation, Yichang 443000, China; 3Changqing Branch of CNPC Logging, Xi’an 710200, China

**Keywords:** core holder, acoustic transducer, anisotropy measurement, rock physical parameter

## Abstract

Many oil and gas fields, especially non-conventional shale and compacted sand reservoirs, have formation anisotropy. The acoustic anisotropy measurement of cores in these reservoirs can guide drilling, well logging, and exploitation. However, almost all core holders are designed for cylinder cores, which are not suitable for all-directional measurements. A three-dimensional measurement device was designed on the basis of the cross-hole sonic logging method. This device mainly consisted of two pairs of transducers, a signal generator, an oscillograph, an omnidirectional positioning system, and a computer control system. By adjusting the measurement latitude and longitude circle automatically, this device scanned spherical sample rocks and obtained full-wave waveforms in all directions. Experiments were performed taking granite from the Jiaodong Peninsula, China, as an example, and the arrival times and velocities of the longitudinal and shear waves were calculated based on the full-wave waveforms. Thereafter, anisotropic physical characterizations were carried out on the basis of these velocities. These data play an important role in guiding formation fracturing and analyzing the stability of borehole walls.

## 1. Introduction

In oil exploration and exploitation, core sampling is often performed on object formations for physical parameter analyses, which can ascertain geological components, rock characteristics, and the processes of geological changes [1,2,3]. Core holders are analysis tools that fix target cores and enable certain kinds of measurements, such as resistivity, permeability, computed tomography, and acoustic velocity measurements. Amongst these measurements, the acoustic parameters of cores can reflect the velocities and amplitudes of longitudinal and shear waves [4]. Then, the lithology, porosity, elasticity, and mechanical characteristics of the cores can be obtained. Since the 1960s, scholars have developed many kinds of devices to obtain the petrophysical parameters of sampling cores through the measurement of acoustic characteristics [5,6]. They discussed the relationships between the acoustic propagation abilities and the physical properties of rocks [7]. Such properties include mechanical parameters, rock structure, mineral composition, and microfractures. These relationships are very important in the exploration and development of oil and gas fields.

Many oil and gas fields, especially non-conventional shale and compacted sand reservoirs, have formation anisotropy [8,9]. The anisotropy measurement of cores in these reservoirs is a challenging issue that can guide drilling, well logging, and exploitation [10]. Given that acoustic signals are highly sensitive to formation anisotropy, acoustic measurement is an effective way to test anisotropy. Almost all core holders are designed for cylinder cores, which are considerably easier to produce and test than other core types. However, cylinder cores are unsuitable for anisotropic measurements [11]. Traditional measurement methods sample and test the same core in different directions to investigate omnidirectional characteristics (Figure 1). However, they cannot obtain the whole three-dimensional (3-D) data, and the cores must be destroyed. In consideration of this issue, a 3-D measurement device is designed to measure the acoustic parameters of anisotropic cores on the basis of the acoustic wave transmission method. By scanning a spherical sample rock in all directions, the full-wave waveforms in all directions are obtained, and the arrival times and velocities of the longitudinal and shear waves are calculated. Thereafter, anisotropic physical characterization can be carried out on the basis of these velocities.

## 2. Design and Test of the 3-D Measurement Device

### 2.1. Structural Design of the Device

The pulse wave reflection and cross-hole sonic logging methods are the two main methods for testing rock elastic velocities. However, if object rocks have the characteristic of strong acoustical attenuation, the pulse wave reflection method obtains low signal–noise ratios. In the cross-hole sonic logging method, transmitters and receivers are fixed on the opposite sides of a sample rock, and the propagation times and lengths of the longitude and shear waves are measured to calculate their velocities. The 3-D measurement device was based on the cross-hole sonic logging method. It could realize high-precision positioning and omnidirectional scanning.

Figure 2 shows that the device mainly consisted of two pairs of transducers [12], a signal generator, an oscillograph, an omnidirectional positioning system [13,14], and a computer control system. The transducers were two pairs of high-precision transmitting and receiving transducers for longitudinal and shear waves, and their domain frequencies were 1 MHz. They were used to carry out transformations between excitation electrical signals and acoustic signals [15]. The signal generator produced pulse electrical signals, which were then amplified by a power preamplifier and loaded on the transmitting transducers. The oscillograph collected weak electronic signals from the receiving transducers [16]. The positioning system mainly consisted of two self-designed transducer holders and a rock sample holder. The transducer holders fixed the transducers, carried out their rotational and positioning motions, and controlled the contact pressure between the transducers and the rock sample with an air pump plunger. The sample holder supported the rock sample with a support rod and performed its rotational motion horizontally. The computer control system exerted overall control and recorded and calculated the collected data.

### 2.2. Workflow of the Measurement Device

A schematic of the measurement conducted by the 3-D measurement device is shown in Figure 3, and the overall workflow was as follows.

Measure the radius of the rock sample and enter this data into the computer control software.Push the rock sample into the instrument and lock it vertically and horizontally. Thereafter, automatically lift the rock sample with the support rod to a suitable position.Place the pair of transducers into the holders and fix them to the initial measurement positions.Produce electrical excitation signals with the signal generator that are amplified by the amplifier and loaded on the transmitting transducers to generate acoustic signals.Propagate the acoustic signals into the rock. These reach the receiving transducers and are then collected by the oscilloscope and uploaded to the computer.Rotate the transducers synchronously in the vertical direction by a latitude of ϕ and repeat steps 3–5 until all the measurements in one circle are finished.Loosen the rock sample, rotate it by a longitude of λ in the horizontal direction, and re-lock it again. Then, repeat steps 3–6 until the measurements are completed in all directions.

The device had the function of automatic measurement. By inputting the latitude step ∆ϕ and the longitude step ∆λ into the computer control system, the device performed measurements automatically to reduce manpower consumption and improve efficiency. 

## 3. Examples of Measurements by the Device

The measurement effect was investigated by taking granite from the Jiaodong Peninsula, China, as an example (Figure 4). The produced spherical rock sample had a black-and-white interlaced laminar structure. The white and black components were quartz and biotite, respectively. The sample had a diameter of 0.30 m and a density of 2850 kg/m^3^. During measurement, the transducers rotated simultaneously to change the latitude angle, and the support rod rotated the rock sample to change the longitude angle.

### 3.1. Measurement of Longitudinal Waves

Firstly, the rock sample was measured by the transducers for longitudinal waves. Vaseline was smeared over the rock sample and the transducers to couple acoustic signals. The working frequency of the transducers was 1 MHz, and the longitude and latitude steps were set to 15° and 10°, respectively. The full-wave received waveforms in different dimensions measured at the longitude of 150° are shown in Figure 5. The velocity and amplitude of the longitude wave were the strongest when the latitude angle was 90° (parallel to the bedding plane). When the latitude angle was changed to the vertical direction of the bedding plane, the velocities of the waves decreased, and their arrival times followed a sinusoidal trend.

The arrival times of the longitudinal waves were calculated on the basis of the scanning-measurement endings. As the acoustic signals were single and clear, the threshold method was utilized to obtain the arrival times of the longitudinal waves. Figure 6 shows the 3-D imaging results, wherein differences in colors indicate differences in arrival times. The arrival times of the acoustic longitudinal waves could be concluded to conform to the same law for different longitude circles. When the connecting directions of the two transducers were parallel to the bedding direction of the rock sample, the arrival time was the shortest, and the propagation speed was the fastest. When the connection direction was perpendicular to the bedding direction, the arrival time was the longest, and the propagation speed was the slowest. 

### 3.2. Measurement of Shear Waves

The velocities of shear waves were measured with a pair of shear wave transducers. Given that shear waves are excited by shearing motions, the receiving transducer must be in the polarization direction of the firing transducer. The conditions and steps of the measurement of shear waves were the same as those for the measurement of longitudinal waves, except that the excitation frequency was set to 0.5 MHz. The 3-D image of the shear wave arrival times is shown in Figure 7. The arrival times of shear waves were significantly later than those of the longitudinal waves shown in Figure 6 because the propagation velocities of shear waves were considerably lower than those of longitudinal waves. Meanwhile, the anisotropy differences followed the same trend: higher in the latitudinal directions and lower in the longitudinal directions.

## 4. Discussion

### 4.1. Calculation of Velocities

Equation (1) could be used to analyze the velocity anisotropy of longitudinal waves in different directions.
(1)$$k\left(\%\right)=\frac{v_{max}-v_{min}}{v_{mean}}\times 10$$
where k is the degree of anisotropy, $v_{max}$ is the maximum velocity, $v_{min}$ is the minimum velocity, and $v_{mean}$ is the average velocity. In accordance with the arrival times, the anisotropies in the longitudinal direction (vertical direction) and latitudinal direction (parallel bedding direction) were calculated and are shown in Figure 8. The anisotropy differences were approximately 3% in the latitudinal direction and 12% in the longitudinal direction. That is, anisotropy differences existed in the vertical direction, which was in line with the bedding structure of the rock sample.

The velocities of the longitudinal and shear waves of the same testing point were combined and fitted. Their relationships are shown in Figure 9. Their velocities were positively related. Their relationship equation is shown in Equation (2).
(2)$$V_{s}=0.3453V_{p}+1562.7$$

The calculated goodness of fit was 0.7416. There were some outside points that were mostly caused by measurement errors, such as the absence of acoustic coupling in these points. Comparison with Ref. [6] shows that the relationship between the compressional and shear wave velocities of the granite samples is similar to those of sandstone and mudstone. 

### 4.2. Calculation of Elastic Physical Parameters

The measurement results could be used to judge the orientation of fractures and to evaluate the brittleness of rocks by inverting their physical parameters because the propagation velocities of acoustic waves in rock are closely related to many physical parameters, such as density, Young’s modulus, Poisson’s ratio, and shear modulus [17]. Therefore, the relationships between the above physical parameters and the velocities of the longitudinal and shear waves were derived.

Young’s modulus:(3)$$E=2G(1+\mu )=2\rho (1+\mu )V_{s}^{2}=\frac{9K\rho V_{s}^{2}}{3K+\rho V_{s}^{2}},$$
where ρ is the density of the rock, K is the bulk modulus, and $V_{s}$ is the velocity of the shear waves.

Shear modulus: (4)$$G=\rho V_{s}^{2}.$$

Bulk modulus:(5)$$K=\rho \left(V_{p}^{2}-\frac{4}{3}V_{s}^{2}\right).$$

Poisson’s ratio:(6)$$\mu =\left[\frac{{\left(\frac{V_{p}}{V_{s}}\right)}^{2}-2}{{\left(\frac{V_{p}}{V_{s}}\right)}^{2}-1}\right].$$

The Poisson’s ratio, bulk modulus, shear modulus, and Young’s modulus values measured in different latitude circles were calculated by substituting the measured velocities of the longitudinal and shear waves into the above formulas and are shown in Figure 10. The above elastic parameters took the maximum values when the latitude angles were approximately from 90° to 100° (parallel to the bedding direction) and the minimum values when the latitude angles were approximately 0° and 160°. This phenomenon also reflected the anisotropic differences in different physical parameters. The data on the sampling core play an important role in guiding formation fracturing and analyzing the stability of borehole walls in the processes of exploration and exploitation.

## 5. Conclusions

A 3-D acoustic measurement device was designed on the basis of the acoustic wave transmission method to measure the anisotropic physical characteristics of formation cores. Its structural design and workflow were introduced. Its measurement effect was investigated by taking granite from the Jiaodong Peninsula, China, as an example. Longitudinal and shear waves were omnidirectionally excited and collected, and their velocities and acoustic anisotropy characteristics were calculated. 

The measured granite rock samples exhibited anisotropy characteristics. The velocities of its longitude and shear waves changed with the measured latitude circle, and the velocities were biggest when the circles were parallel to the bedding direction of the rock sample. The anisotropy characteristics were very small in the longitude orientations, and little velocity change happened.The velocities of the longitudinal and shear waves were positively related. The calculated goodness of fit was 0.7416. The relationship between the compressional and shear wave velocities of the granite samples was similar to those of sandstone and mudstone.The anisotropy characteristics of Poisson’s ratio, bulk modulus, shear modulus, and Young’s modulus were calculated based on the measured velocities of the longitudinal and shear waves. These data play an important role in guiding formation fracturing and analyzing the stability of borehole walls in the processes of exploration and exploitation.The design principle of the 3-D acoustic measurement device could also be used in devices for other kinds of measurements, such as resistivity, permeability, and computed tomography measurements.

## Figures and Tables

**Figure 1 sensors-22-09473-f001:**
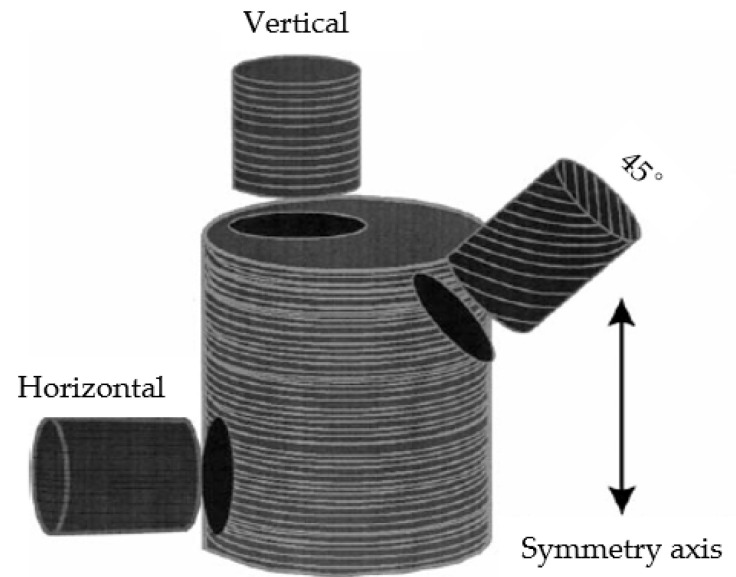
Traditional measurement methods sample the same core in different directions.

**Figure 2 sensors-22-09473-f002:**
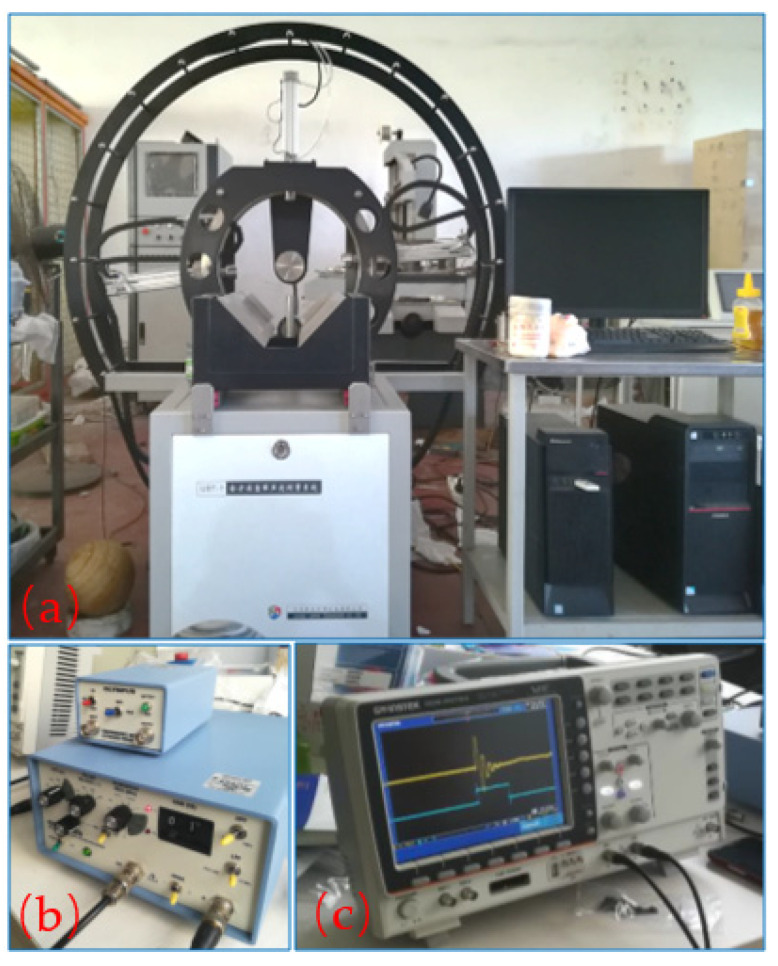
Actual picture of the three-dimensional anisotropic measurement device: (**a**) omnidirectional positioning system, (**b**) signal generator and amplifier, and (**c**) oscillograph.

**Figure 3 sensors-22-09473-f003:**
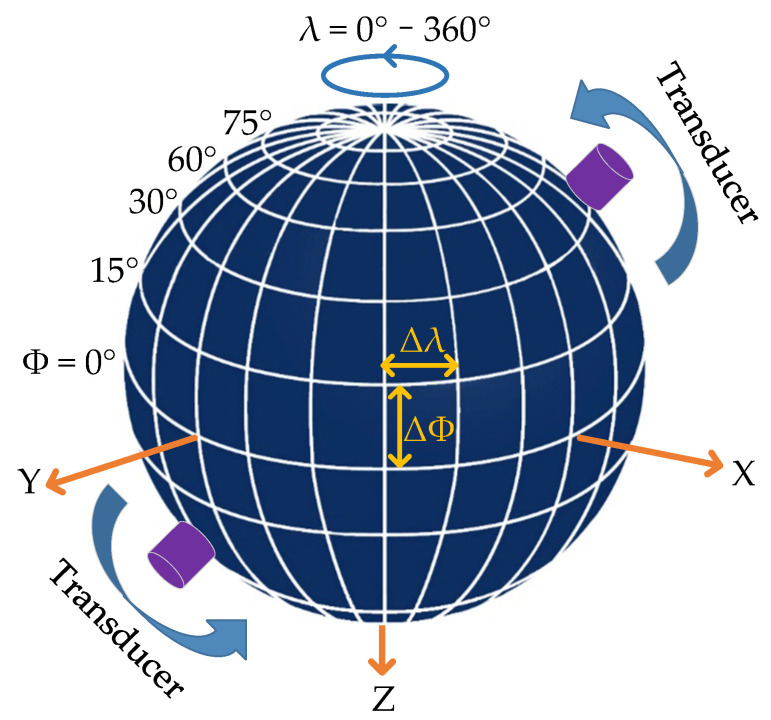
Schematic of measurements conducted by the 3-D measurement device.

**Figure 4 sensors-22-09473-f004:**
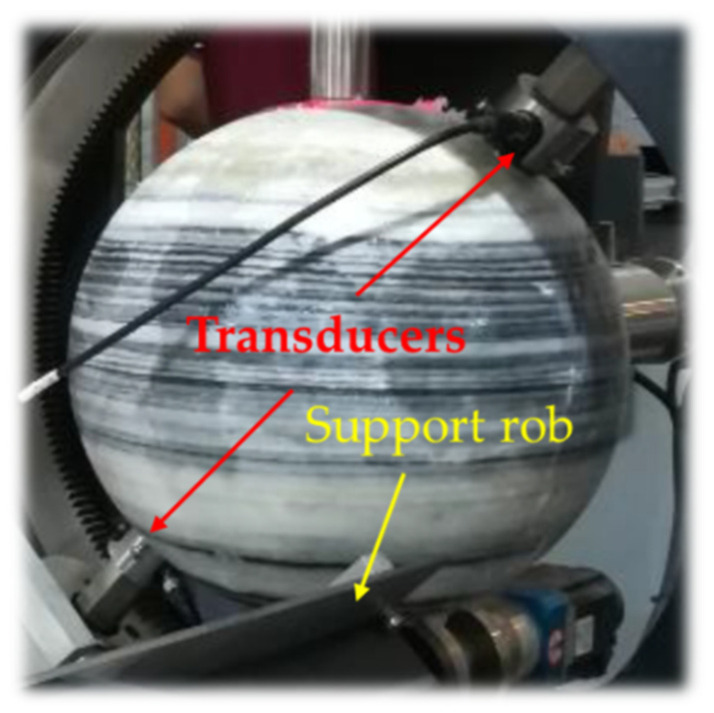
Granite rock sample from Jiaodong Peninsula, China.

**Figure 5 sensors-22-09473-f005:**
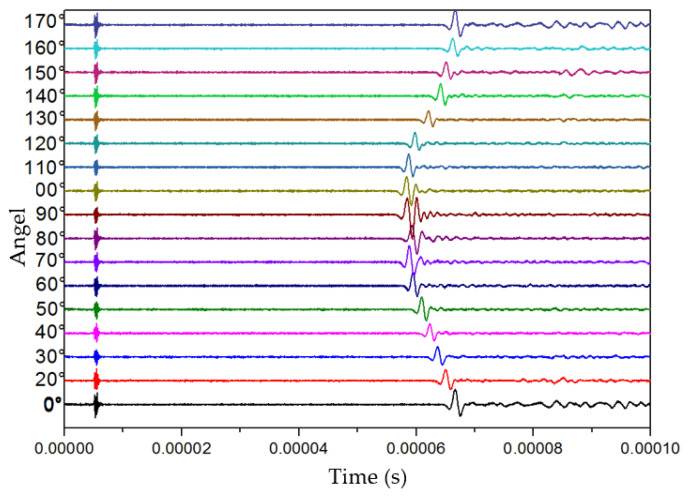
Full-wave received waveforms in different dimensions measured at the longitude of 150°.

**Figure 6 sensors-22-09473-f006:**
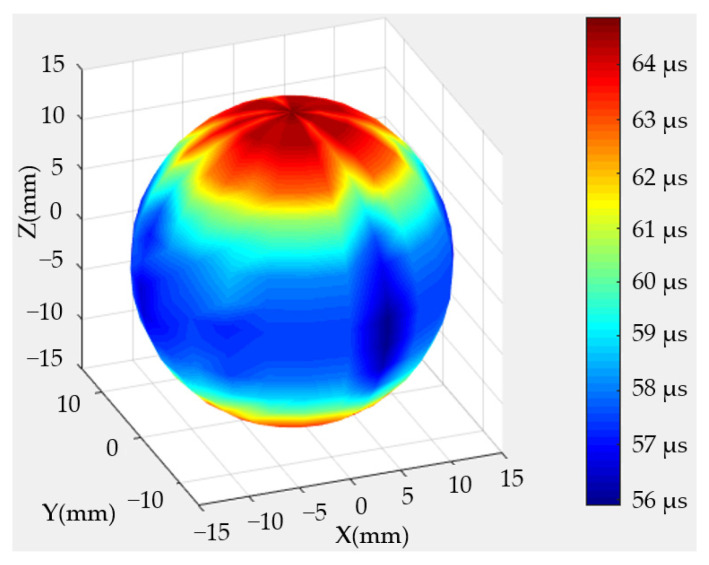
Arrival times of longitudinal waves on the basis of scanning-measurement endings.

**Figure 7 sensors-22-09473-f007:**
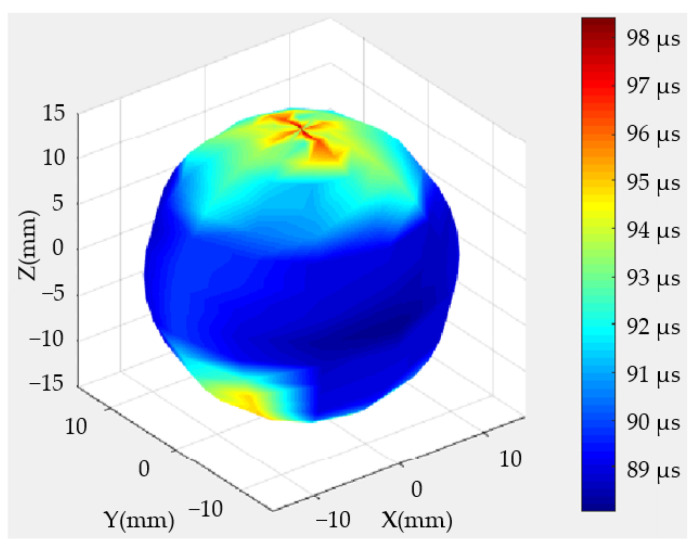
Arrival times of shear waves on the basis of scanning-measurement endings.

**Figure 8 sensors-22-09473-f008:**
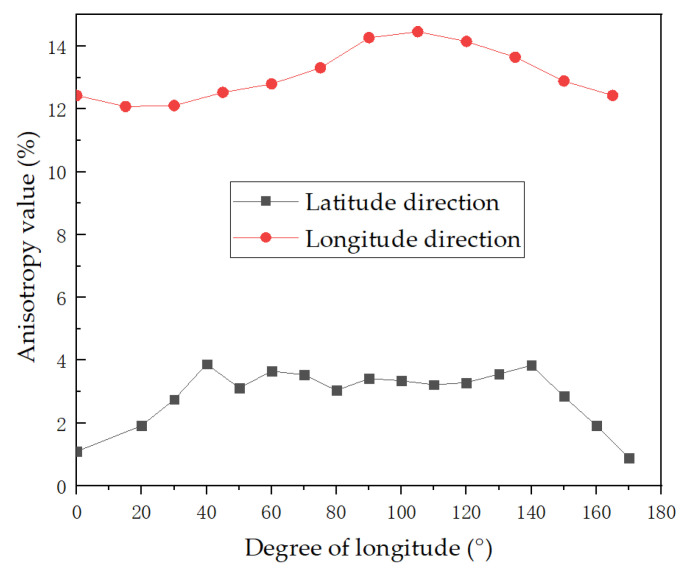
Velocity anisotropies in the longitudinal direction and latitudinal direction.

**Figure 9 sensors-22-09473-f009:**
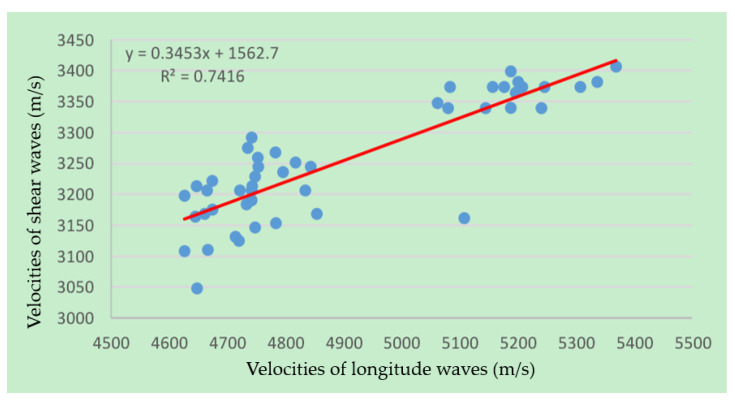
Measured velocities of the longitudinal and shear waves of the same testing point (blue dots) and their fit curve (red line).

**Figure 10 sensors-22-09473-f010:**
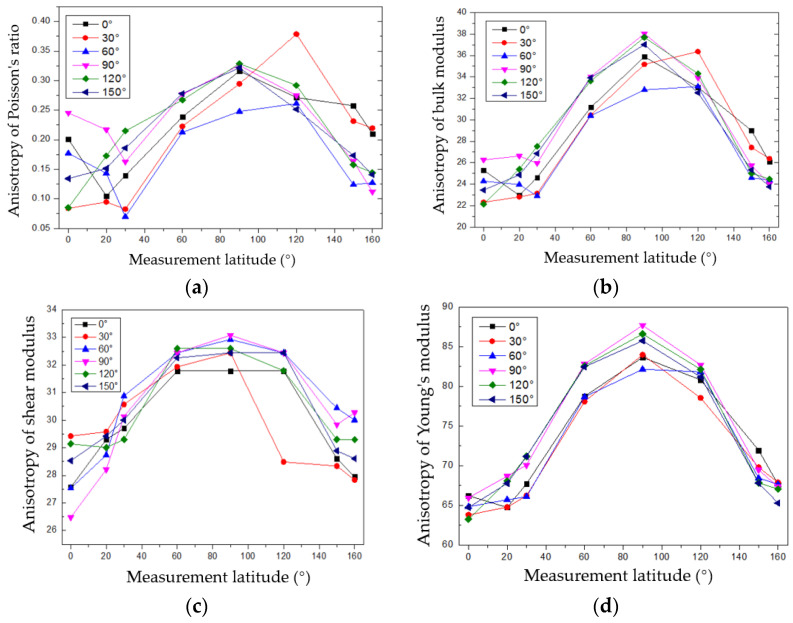
Calculated elastic anisotropic parameters: (**a**) Poisson’s ratio, (**b**) bulk modulus, (**c**) shear modulus, and (**d**) Young’s modulus.

## Data Availability

Not applicable.

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
