# Peer review of "Three-Dimensional Acoustic Device for Testing the All-Directional Anisotropic Characteristics of Rock Samples"

_sensors, 2022, doi:10.3390/s22239473_

Round 1
Reviewer 1 Report
Interesting paper in which the authors propose a three-dimensional measurement device was designed on the basis of the cross-hole sonic logging method. For obtaining the full-wave wave forms in all directions, including the arrival times and velocities of the longitudinal and shear. Thus, the anisotropic physical characterizations of the material are obtaining. The subject is interesinting and the producto can be useful, the paper is well written and structured, so it should be accepted after minor changes.
1) The authors must explain how the arrival time is obtained. Threshold method? Akaike method?
2) What happens if longitudinal and shear waves are mixed? How do you determine the arrival time in this case?
3) Figure 9: There are outsider points. Why?
Reviewer 2 Report
You have provided general information in the abstract section. Please give a summary of your work in the abstract section. Although your work is experimental, you did not discuss them sufficiently in the result section. Also, is it necessary to change the 2nd and 3rd part as the title? because the result should be considered as part of discussion and conclusion. There are experimental studies done in the article. but not enough discussion and conclusions have been written. I also found that the authors cited too much. Please reduce them.
Although your article is a good experimental work, it has not been adequately presented.
Round 2
Reviewer 2 Report
Check english grammar